# FORMAL LANGUAGE CONSTRAINED MARKOV DECISION PROCESSES

## ABSTRACT

In order to satisfy safety conditions, an agent may be constrained from acting freely. A safe controller can be designed a priori if an environment is well understood, but not when learning is employed. In particular, reinforcement learned (RL) controllers require exploration, which can be hazardous in safety critical situations. We study the benefits of giving structure to the constraints of a constrained Markov decision process by specifying them in formal languages as a step towards using safety methods from software engineering and controller synthesis. We instantiate these constraints as finite automata to efficiently recognise constraint violations. Constraint states are then used to augment the underlying MDP state and to learn a dense cost function, easing the problem of quickly learning joint MDP/constraint dynamics. We empirically evaluate the effect of these methods on training a variety of RL algorithms over several constraints specified in Safety Gym, MuJoCo, and Atari environments.

## 1 INTRODUCTION

The ability to impose safety constraints on an agent is key to the deployment of reinforcement learning (RL) systems in real-world environments (Amodei et al., 2016). Controllers that are derived mathematically typically rely on a full a priori analysis of agent behavior remaining within a pre-defined envelope of safety in order to guarantee safe operation (Aréchiga & Krogh, 2014). This approach restricts controllers to pre-defined, analytical operational limits, but allows for verification of safety properties (Huth & Kwiatkowska, 1997) and satisfaction of software contracts (Helm et al., 1990), which enables their use as a component in larger systems. By contrast, RL controllers are free to learn control trajectories that better suit their tasks and goals; however, understanding and verifying their safety properties is challenging. A particular hazard of learning an RL controller is the requirement of exploration in an unknown environment. It is desirable not only to obey constraints in the final policy, but also throughout the exploration and learning process (Ray et al., 2019).

The goal of safe operation as an optimization objective is formalized by the constrained Markov decision process (CMDP) (Altman, 1999), which adds to a Markov decision process (MDP) a cost signal similar to the reward signal, and poses a constrained optimization problem in which discounted reward is maximized while the total cost must remain below a pre-specified limit per constraint. We use this framework and propose specifying CMDP constraints in formal languages to add useful structure based on expert knowledge, e.g., building sensitivity to proximity into constraints on object collision or converting a non-Markovian constraint into a Markovian one (De Giacomo et al., 2020).

A significant advantage of specifying constraints with formal languages is that they already form a well-developed basis for components of safety-critical systems (Huth & Kwiatkowska, 1997; Clarke et al., 2001; Kwiatkowska et al., 2002; Baier et al., 2003) and safety properties specified in formal languages can be verified a priori (Kupferman et al., 2000; Bouajjani et al., 1997). Moreover, the recognition problem for many classes of formal languages imposes modest computational requirements, making them suitable for efficient runtime verification (Chen & Roşu, 2007). This allows for low-overhead incorporation of potentially complex constraints into RL training and deployment.

We propose (1) a method for posing formal language constraints defined over MDP trajectories as CMDP cost functions; (2) augmenting MDP state with constraint automaton state to more explicitly encourage learning of joint MDP/constraint dynamics; (3) a method for learning a dense cost function given a sparse cost function from joint MDP/constraint dynamics; and (4) a method based on

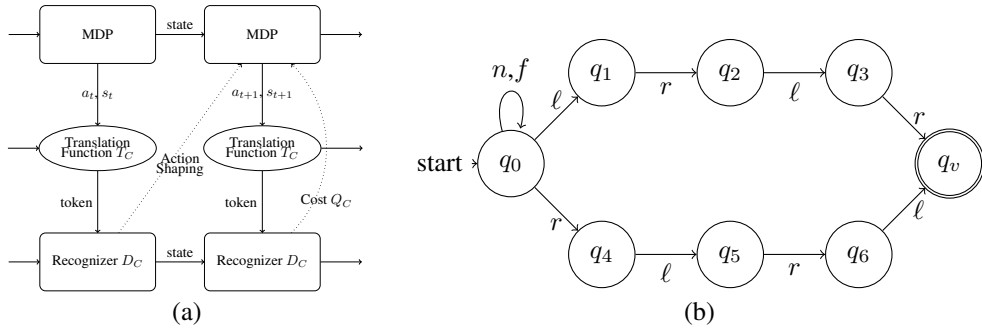

Figure 1: (a) Illustration of the formal language constraint framework operating through time. State is carried forward through time by both the MDP and the recognizer, $D_C$. (b) No-1D-dithering constraint employed in the Atari and MuJoCo domains: $.^* (\ell\, r)^2 | (r\, \ell)^2$ (note, all unrepresented transitions return to $q_0$).

constraint structure to dynamically modify the set of available actions to guarantee the prevention of constraint violations. We validate our methods over a variety of RL algorithms with standard constraints in Safety Gym and hand-built constraints in MuJoCo and Atari environments.

The remainder of this work is organized as follows. Section 2 presents related work in CMDPs, using expert advice in RL and safety, as well as formal languages in similar settings. Section 3 describes our definition of a formal language-based cost function, as well as how it's employed in state augmentation, cost shaping, and action shaping. Section 4 details our experimental setup and results and finally, discussion of limitations and future work are located in Section 5.

## 2  RELATED WORK

**Safety and CMDP Framework**    The CMDP framework doesn't prescribe the exact form of constraints or how to satisfy the constrained optimization problem. Chow et al. (2017) propose conditional value-at-risk of accumulated cost and chance constraints as the values to be constrained and use a Lagrangian formulation to derive a Bellman optimality condition. Dalal et al. (2018) use a different constraint for each MDP state and a safety layer that analytically solves a linearized action correction formulation per state. Similarly, Pham et al. (2018) introduce a layer that corrects the output of a policy to respect constraints on the dynamic of a robotic arm.

**Teacher Advice**    A subset of work in safe exploration uses expert advice with potential-based reward shaping mechanisms (Ng et al., 1999). Wiewiora et al. (2003) introduce a general method for incorporating arbitrary advice into the reward structure. Saunders et al. (2017) use a human in the loop to learn an effective RL agent while minimizing cost accumulated over training. Camacho et al. (2017a;b) use DFAs with static reward shaping attached to states to express non-Markovian rewards. We generalise this work with a learned shaping function in the case of dense soft constraints, and by generalising from reward shaping to other CMDP learning mechanisms. Similar to teacher advice is shielding (Jansen et al., 2018; Alshiekh et al., 2018), in which an agent's actions are filtered through a shield that blocks actions that would introduce an unsafe state (similar to hard constraints; section 3), but typically requires MDP states to be enumerable and few enough that a shield can be constructed efficiently.

**Formal Languages**    Formal languages and automata have been used before in RL for task specification or as task abstractions (options) in hierarchical reinforcement learning (Icarte et al., 2018b; Li et al., 2017; Wen et al., 2017; Mousavi et al., 2014). In some cases, these automata were derived from Linear Temporal Logic (LTL) formulae, in others LTL or other formal language formulae have been directly used to specify tasks (Icarte et al., 2018a). Littman et al. (2017) defines a modified LTL designed for use in reinforcement learning. In robotics, LTL is used for task learning (Li et al., 2017), sometimes in conjunction with teacher demonstrations (Li et al., 2018). Zhu et al. (2019) and Fulton & Platzer (2019) both study the use of formal languages for safe RL, though each makes assumptions about a prior knowledge of the environment dynamics. Hasanbeig et al. (2018) and Hasanbeig et al. (2020) learn a product MDP with a safety constraint specified in a formal language,

but require knowledge of MDP structure and don't scale to large MDPs. Hahn et al. (2019) uses an omega-regular language to specify an objective and constructs a product MDP to ensure the objective is reached almost surely, but requires large reward shaping which prevents learning when used for a constraint rather than objective.

## 3    FORMAL LANGUAGE CONSTRAINED MDPs

The constrained Markov decision process (CMDP) (Altman, 1999) extends the Markov decision process (Sutton & Barto, 2018) to incorporate constraints. The difference is an additional set of cost functions $c_i : S \times A \times S \to \mathbb{R}$ and set of cost limits $d_i \in \mathbb{R}$. Then, the constrained optimization problem is $\arg\max_\pi J_r(\pi)$ s.t. $J_{c_i}(\pi) \leq d_i, i = 1, \ldots, k$ where $J_r(\pi)$ is a return-based objective, e.g., finite horizon discounted return defined $J_r(\pi) = \mathbb{E}_{\tau \sim \pi}[\sum_{t \in \tau} \gamma^t r_t]$ and $J_{c_i}$ is a cost-based constraint function defined similarly, replacing $r_t$ with $c_{i,t}$.

We propose formal language constrained MDPs (FLCMDPs) as a subset of CMDPs in which each constraint $C_i \subset (S \times A \times S)^*$ is defined by a set of prohibited trajectories. (Subscript $i$ is suppressed from this point without loss of generality). Because $C$ is defined by a formal language, it can be recognized efficiently by an automaton, which we use to construct the cost function. We define three functions for interacting with the constraint automaton: a translation function $T_C : (S \times A \times S) \to \Sigma_C$ that converts MDP transitions into a symbol in the recognizer's input language, a recognizer function $D_C : Q_c \times \Sigma_C \to Q_C$ that steps the recognizer automaton using its current state and the input symbol and returns its next state, and finally a cost assignment $G_C : Q_C \to \mathbb{R}$ that assigns a real-valued cost to each recognizer state. The composition of these three functions forms a CMDP cost function defined $c = G_C \circ D_C \circ T_C : (S \times A \times S) \times Q_c \to \mathbb{R}$, which requires constraint state $Q_c$ to be tracked along with the MDP state, $S$. The interaction of these functions with the underlying MDP framework is illustrated in Figure 1(a), where the constraint uses the MDP state and action at time $t$ to calculate the cost signal at time $t$ and, if action shaping is being employed as discussed below, influence the action at time $t + 1$. We note that this construction does not require knowledge of the MDP transition probabilities and further, its complexity is independent of the size of the MDP, which allows it to scale to large, challenging environments.

**Translation Function** $T_C$    The translation function accepts as input the MDP state and action at each time step, and outputs a token in the discrete, finite language of the associated recognizer. This allows the recognizer automaton to be defined in a small, discrete language, rather than over unwieldy and potentially non-discrete MDP transitions. Further, freedom in choice of input language allows for flexible design of the constraint automaton to encode the desired inductive bias, and thus more meaningful structured states.

**Recognizer Function** $D_C$    Each constraint is instantiated with a finite automaton recognizer that decides whether a trajectory is in the constraint set. The only necessary assumption about the recognizer is that it defines some meaningful state that may be used for learning the constraint dynamics. Our implementation uses a deterministic finite automaton (DFA) as the recognizer for each constraint, defined as $(Q, \Sigma, \delta, q_0, F)$, where $Q$ is the set of the DFA's states, $\Sigma$ is the alphabet over which the constraint is defined, $\delta : Q \times \Sigma \to Q$ is the transition function, $q_0 \in Q$ is the start state, and $F \subset Q$ is the set of accepting states that represent constraint violations. The DFA is set to its initial state at the start of each episode and is advanced at each time step with the token output by the translation layer. Although our experiments use DFAs as a relatively simple recognizer, the framework can be easily modified to work with automata that encode richer formal languages like pushdown automata or hybrid automata.

**Constraint State Augmentation**    In order to more efficiently learn constraint dynamics, the MDP state $s_t$ is augmented with a one-hot representation of the recognizer state $q_t$. To preserve the Markov property of the underlying MDP, state augmentation should contain sufficient information about the recognizer state and, if it is stateful, the translation function. To enhance performance, the one-hot state is embedded to $\lceil \log_2(|Q|) \rceil$ dimensions before being input into any network and the embedding is learned with gradients backpropagated through the full network.

**Cost Assignment Function** The cost assignment function $G_C$ assigns a real-valued cost to each state of the recognizer. This cost can be used in optimization to enforce the constraint with a Lagrangian-derived objective penalty, or via reward shaping, which updates the reward function to $r_t - \lambda c_t$, where $\lambda$ is a scaling coeffficient.

Cost assignments are frequently sparse, where $G_C$ is only non-zero at accepting states that recognize a constraint violation. This poses a learning problem for optimization-based methods that use reward shaping or an objective penalty to solve the CMDP. A goal of constrained RL is to minimize accumulated constraint violations over training but, to ensure that the frequency of violations is small, the optimization penalty can be large relative to the reward signal. This can lead to a situation in which an unnecessarily conservative policy is adopted early in training, slowing exploration. We next propose a method for learning a dense cost function that takes advantage of the structure of the constraint automaton to more quickly learn constraint dynamics and avoid unnecessarily conservative behavior.

**Learned Dense Cost** The goal of learning a dense cost is not to change the optimality or near-optimality of a policy with respect to the constrained learning problem. Thus, we use the form of potential-based shaping: $F(s_{t-1}, a_t, s_t) = \gamma \Phi(s_t) - \Phi(s_{t-1})$, where $\Phi$ is a potential function (see Ng et al. (1999) for details). This is added as a shaping term to the sparse cost to get the dense cost $G'_C(q_{t-1}, q_t) = G_C(q_t) + \beta(\gamma \Phi(q_t) - \Phi(q_{t-1}))$ , where $\beta$ scales the added dense cost relative to the sparse cost, and $\Phi$ is a function of the recognizer state rather than the MDP state, which requires $s_{t-2}$ and $a_{t-2}$ as additional inputs to calculate $q_{t-1}$. Generally, if the value of $\Phi$ increases as the automaton state is nearer to a violation, then the added shaping terms add cost for moving nearer to a constraint violation and refund cost for backing away from a potential violation.

In our experiments, the potential $\Phi^\pi(q_t)$ is defined using $t_v(q_t)$, which is a random variable defined as the number of steps between visiting recognizer state $q_t$ and an accepting recognizer state. This variable's distribution is based on $\pi$ and the MDP's transition function. Its value is small if a violation is expected to occur soon after reaching $q_t$ and vice-versa. We then define the potential function as $\Phi^\pi(q_t) = \left(\frac{1}{2}\right)^{(\mathbb{E}_\pi[T_v(q_t)]/t_v^{baseline})}$ , which ensures that its value is always in $[0, 1]$ and rises exponentially as the expected time to a violation becomes smaller. If the expected time to next violation is much larger than the provided baseline, $t_v^{baseline}$, then the potential value becomes small, as shaping is unnecessary in safe states. The expected value of $T_v(q_t)$ over trajectories may be calculated as a Monte Carlo estimate from rollouts by counting the number of steps between each visit to $q_t$ and the next constraint violation. Each $T_v$ is tracked with an exponential moving average and is updated between episodes to ensure that it's stationary in each rollout. We set $t_v^{baseline}$ to be the ratio of estimated or exact length of an episode and the constraint limit $d_i$, but find empirically that the the method is resilient to the exact choice.

**Hard Constraints and Action Shaping** When safety constraints are strict, i.e., when the limit on the number of constraint violations $d$ is zero, the set of available actions is reduced to ensure a violation cannot occur. If a constraint isn't fully state-dependent (i.e., there is always an action choice that avoids violation), then action shaping can guarantee that a constraint is never violated. Otherwise, knowledge of which actions lead to future violating trajectories requires knowledge of the underlying MDP dynamics, which is possible by learning a model that converts state constraints into state-conditional action constraints as in Dalal et al. (2018).

Our implementation of hard constraints initially allows the agent to freely choose its action, but before finalizing that choice, simulates stepping the DFA with the resulting token from the translation function and, if that lookahead step would move it into a violating state, it switches to the next best choice until a non-violating action is found. For the constraints in our experiments, it is always possible to choose a non-violating action. A known safe fallback policy can be employed in the case when an episode cannot be terminated. Action shaping can be applied during training or deployment, as opposed to reward shaping, which is only applied during training. We experiment with applying action shaping only during training, only during evaluation, or in both training and evaluation.

## 4    EXPERIMENTAL EVALUATION

### 4.1    CONSTRAINTS

We evaluated FLCMDPs on four families of constraints, which we define with regular expressions. The cost function defined by each constraint is binary, unless we use dense cost shaping.

**No-dithering:** A no-dithering constraint prohibits movements in small, tight patterns that cover very small areas. In one dimension, we define dithering as actions are taken to move left, right, left, and right in order or the opposite, i.e., $.^* (\ell\, r)^2 | (r\, \ell)^2$. The automaton encoding this constraint is depicted in Figure 1(b). In environments with two-dimensional action spaces, such as Atari Seaquest, we generalize this to vertical and diagonal moves and constrains actions that take the agent back to where it started in at most four steps[1]. In MuJoCo, constraints are applied per joint and the translation function maps negative and positive-valued actions to '$\ell$' and '$r$', respectively.

**No-overactuating:** A no-overactuating constraint prohibits repeated movements in the same direction over a long period of time. In Atari environments, this forbids moving in the same direction four times in a row, i.e., $.^* (\ell^4 \cup r^4)$. In two dimensions, this is extended to include moving vertically: $.^* (L^4 \cup R^4 \cup U^4 \cup D^4)$. Each of the left ($L$), right ($R$), up ($U$) and down ($D$) tokens is produced by the translation function from the primary direction it's named after or diagonal moves that contain the primary direction, e.g., $L = \ell \cup \ell+u \cup \ell+d$, where "$\ell+u$" is the atomic left-up diagonal action. In MuJoCo environments, overactuation is modelled as occurring when the sum of the magnitudes of joint actuations exceeds a threshold. This requires the translation function to discretize the magnitude in order for a DFA to calculate an approximate sum. The MDP state-based version is "dynamic actuation", which sets the threshold dynamically based on a discretized distance from the goal.

**Proximity:** The proximity constraint, used in Safety Gym, encodes the distance to a collision with any of the variety of hazards found in its environments. The translation function uses the max value over all the hazard lidars, which have higher value as a hazard comes closer, and discretizes it into one of ten values. The constraint is defined as being violated if the agent contacts the hazard, which is identical to the constraint defined in the Safety Gym environments and described in Ray et al. (2019).

**Domain-specific:** In addition to the previously described simple constraints, we define hand-built constraints for the Breakout and Space Invaders Atari environments. These constraints are designed to mimic specific human strategies in each environment for avoiding taking actions that end the episode. In Atari Breakout, we define the "paddle-ball" constraint, which limits the allowed horizontal distance between the ball and the center of the paddle. In Atari Space Invaders, we define the "danger zone" constraint, which puts a floor on the the allowed distance between the player's ship and the bullets fired by enemies. We provide more details of each constraint in Appendix B.

### 4.2    ENVIRONMENTS

In Safety Gym, the Spinning Up implementation of PPO with Lagrangian optimization penalization was employed, with hyperparameters as chosen identically to Ray et al. (2019). We modified each network to concatenate the constraint state augmentation with the input and used $d = 25$ for the expected cost limit. All safety requirements are accounted for in a single constraint and we report the constraint violations as accounted for in the Safety Gym environments rather than as reported by the finite automaton (though these are identical when not using cost shaping). Each environment, which is randomly re-arranged each episode, is made up of a pairing of a robot and a task. The robots are Point, which turns and moves, and Car, which is wheeled with differential drive control. The tasks are Goal, which requires moving into a goal area, Button, which requires pressing a series of buttons, and Push, which requires moving a box into a goal area. More details can be found in Ray et al. (2019).

In Atari environments (Bellemare et al., 2013), we modified the DQN implemented in OpenAI Baselines (Dhariwal et al., 2017) by appending the state augmentation to the output of its final convolutional layer. Reward shaping was used for soft constraint enforcement with the penalty fixed at one of $\{0, -0.001, -0.0025, -0.005, -0.01\}$, and each agent was trained for 10M steps before collecting data from an evaluation phase of 100K steps for 15 or more train/eval seed pairs for each hyperparameter combination. For MuJoCo environments (Brockman et al., 2016), we trained the

---

[1]The regex describing this constraint is included in Appendix D.

Table 1: Metrics averaged over the last 25 episodes of training in Safety Gym environments with PPO-Lagrangian methods, normalized relative to unconstrained PPO metrics. Cost rate is the accumulated cost regret over the entirety of training.

| Environment | FLCMDP State Augmented | | | Baseline | | |
|---|---|---|---|---|---|---|
| | Return | Violation | Cost Rate | Return | Violation | Cost Rate |
| Point-Goal1 | 0.750 | **0.427** | **0.281** | **0.918** | 0.925 | 0.503 |
| Point-Goal2 | **0.195** | 0.083 | **0.078** | 0.021 | **0.062** | 0.155 |
| Point-Button1 | 0.252 | **0.129** | **0.128** | **0.343** | 0.296 | 0.218 |
| Point-Button2 | **0.251** | **0.130** | 0.141 | 0.166 | 0.255 | **0.118** |
| Point-Push1 | 0.549 | **0.042** | **0.061** | **0.692** | 0.496 | 0.543 |
| Point-Push2 | **0.938** | 0.173 | 0.148 | 0.670 | 0.295 | 0.258 |
| Car-Goal1 | **0.825** | 0.295 | 0.284 | 0.803 | 0.475 | 0.445 |
| Car-Goal2 | 0.005 | **0.011** | 0.079 | **0.021** | 0.046 | 0.108 |
| Car-Button1 | **0.022** | 0.083 | 0.071 | 0.018 | **0.039** | 0.118 |
| Car-Button2 | **0.031** | 0.147 | 0.076 | 0.009 | **0.009** | 0.078 |
| Car-Push1 | 0.737 | **0.032** | **0.069** | **0.882** | 0.387 | 0.420 |
| Car-Push2 | **0.256** | 0.086 | 0.124 | 0.025 | 0.115 | 0.202 |

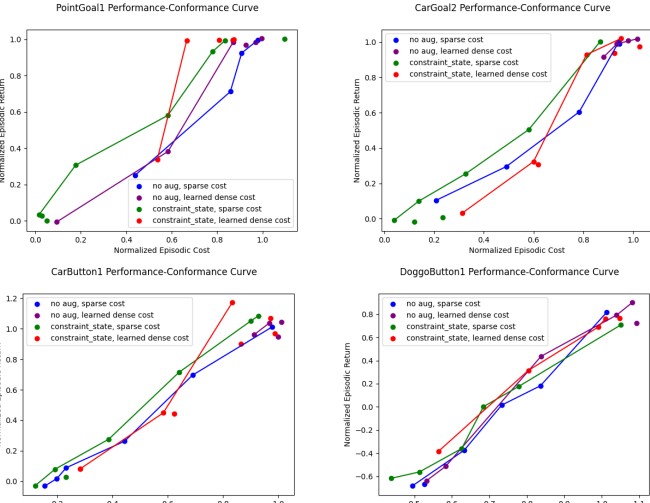

Figure 2: Performance/conformance curves in selected Safety Gym environments with Pareto frontiers plotted per reward shaping method. We observe that using state augmentation (green) consistently outperforms the baseline (blue) at all levels of reward shaping, which are anti-correlated with episodic cost and episodic return. The use of cost shaping (purple) produces gains in return at a given amount of cost only at small reward shaping values correlating to high return and cost. Consequently, the combination of state augmentation and cost shaping inherits this behavior of being more effectiveness when cost/return are higher. The full set of plots is included in Appendix C.

Baselines PPO agent (Schulman et al., 2017) with dense constraints and the state augmentation concatenated to the input MDP observation. Reward shaping similar to Atari was employed for constraint enforcement. Atari and MuJoCo environments do not have constraints built in, so we report the number of constraint violations per episode from the custom constraints and minimize them without a specific goal value.

## 4.3 RESULTS

We experiment with our methods to evaluate the usefulness of formal language constraints in optimizing three objectives. In the final policy output by the training process, it is desirable to simultaneously maximize return per episode and minimize constraint violations per episode, or keep them below the specified limit. The third objective is to minimize accumulated cost regret

Table 2: Atari reward shaping with state augmentation, choosing hyperparameters that minimize constraint violations per episode. "Dense" refers to whether the dense cost term was used and "reward shaping" refers to the fixed reward shaping coefficient $\lambda$.

| | | | | FLCMDP State Augmented | | | Baseline | | |
|---|---|---|---|---|---|---|---|---|---|
| Environment | Constraint | Dense | Reward Shaping | Mean Episode Reward | Mean Step Reward | Mean Viols/100 steps | Mean Episode Reward | Mean Step Reward | Mean Viols/100 steps |
| Breakout | actuation | False | −0.001 | **297.12 ± 8.07** | **0.15 ± 0.0039** | **0.45 ± 0.00067** | 272.19 ± 43.12 | 0.14 ± 0.01 | 13.59 ± 0.025 |
| | dithering | False | −0.001 | 263.57 ± 11.14 | **0.15 ± 0.0068** | **0.0008 ± 8.4e-06** | 272.19 ± 43.12 | 0.14 ± 0.01 | 0.12 ± 0.001 |
| | paddle ball | True | −0.0025 | **314.79 ± 15.09** | **0.17 ± 0.0056** | 6.15 ± 0.0031 | 272.19 ± 43.12 | 0.14 ± 0.01 | 13.40 ± 0.022 |
| Seaquest | actuation | False | −0.01 | 1858.65 ± 478.56 | 0.76 ± 0.18 | **2.71 ± 0.0066** | 2250.13 ± 647.92 | **0.96 ± 0.21** | 9.74 ± 0.017 |
| | dithering | False | −0.01 | 1608.66 ± 41.25 | 0.75 ± 0.013 | **0.081 ± 0.001** | 2250.13 ± 647.92 | **0.96 ± 0.21** | 1.61 ± 0.007 |
| SpaceInvaders | actuation | False | −0.01 | 598.78 ± 39.98 | **0.63 ± 0.033** | 5.39 ± 0.017 | 604.86 ± 44.86 | 0.62 ± 0.04 | 10.88 ± 0.011 |
| | dangerzone | True | −0.005 | **629.32 ± 28.72** | **0.65 ± 0.017** | **0.00 ± 0.00** | 604.86 ± 44.86 | 0.62 ± 0.04 | **0.00 ± 0.00** |
| | dithering | True | −0.01 | 595.25 ± 20.25 | **0.63 ± 0.021** | **0.00 ± 0.00** | 604.86 ± 44.86 | 0.62 ± 0.04 | 0.53 ± 0.0064 |

Table 3: Atari reward shaping with state augmentation, choosing hyperparameters that maximize cumulative reward per episode. "Dense" refers to whether the dense cost term was used and "reward shaping" refers to the fixed reward shaping coefficient $\lambda$.

| | | | | FLCMDP State Augmented | | | Baseline | | |
|---|---|---|---|---|---|---|---|---|---|
| Environment | Constraint | Dense | Reward Shaping | Mean Episode Reward | Mean Step Reward | Mean Viols/100 Steps | Mean Episode Reward | Mean Step Reward | Mean Viols/100 Steps |
| Breakout | actuation | False | −0.001 | **297.12 ± 8.07** | **0.15 ± 0.0039** | **0.45 ± 0.00067** | 272.19 ± 43.12 | 0.14 ± 0.01 | 13.59 ± 0.025 |
| | dithering | True | −0.005 | **302.24 ± 43.81** | 0.14 ± 0.02 | **0.11 ± 0.001** | 272.19 ± 43.12 | **0.14 ± 0.01** | 0.12 ± 0.001 |
| | paddle ball | True | −0.0025 | **314.79 ± 15.09** | **0.17 ± 0.0056** | 6.15 ± 0.0031 | 272.19 ± 43.12 | 0.14 ± 0.01 | 13.40 ± 0.022 |
| Seaquest | actuation | True | −0.0025 | 2339.54 ± 442.02 | 0.93 ± 0.11 | **4.43 ± 0.016** | 2250.13 ± 647.92 | **0.96 ± 0.21** | 9.74 ± 0.017 |
| | dithering | False | −0.001 | 1997.91 ± 539.75 | 0.86 ± 0.23 | **1.58 ± 0.011** | 2250.13 ± 647.92 | **0.96 ± 0.21** | 1.61 ± 0.007 |
| SpaceInvaders | actuation | False | −0.005 | **646.99 ± 50.55** | **0.64 ± 0.04** | 29 ± 0.0053 | 604.86 ± 44.86 | 0.62 ± 0.04 | **10.88 ± 0.011** |
| | dangerzone | False | −0.001 | **687.37 ± 16.75** | **0.63 ± 0.01** | **0.00 ± 0.00** | 604.86 ± 44.86 | 0.62 ± 0.04 | **0.00 ± 0.00** |
| | dithering | False | −0.001 | **640.35 ± 25.94** | **0.67 ± 0.09** | **0.17 ± 0.00027** | 604.86 ± 44.86 | 0.62 ± 0.04 | 0.53 ± 0.0064 |

over the course of training. To examine the proposed methods, we investigate two questions. First, what effect do the proposed methods have on accumulated regret? In Section 4.3.1, we compare the proposed methods against a baseline when combined with PPO using a Lagrangian approach in Safety Gym (Ray et al., 2019). Second, how should hyperparameters be chosen to minimize or maximize each objective respectively? In Section 4.3.2 we examine which hyperparameter choices worked well in various Atari and MuJoCo environments using reward shaping. Finally, in Section 4.3.3 we see what effect enforcing zero constraint violations with action shaping has on Atari environments.

### 4.3.1 LAGRANGIAN RESULTS AND ACCUMULATED COST

Table 1 compares PPO with Lagrangian constraint enforcement with and without constraint state augmentation. The clearest trend is in the reduction of the cost rate, which measures accumulated cost regret, often by between almost one half and an order of magnitude. This results from the inclusion of the helpful inductive bias provided by the constraint structure. This result is not surprising, but does quantify the magnitude of the benefit that a low-overhead method like formal language constraints can have. Qualitatively, we noted that the earliest steps of training had decreased performance generally as the embedding of the constraint state was being learned, but quickly surpassed baseline performance once the updates of the embedded representation became small.

Ray et al. (2019) says that algorithm $A_1$ dominates $A_2$ when they are evaluated under the same conditions, the cost and return of $A_1$ are at least as good as that of $A_2$, and at least one of cost and return is strictly better for $A_1$. By this definition, the state augmented approach strictly dominates the baseline in 6 of 12 environments, while coming close in most of the rest. Specifically, we also note that state augmentation allowed a significant step to be taken in closing the gap between unconstrained PPO return and PPO-Lagrangian in the Point-Goal2 and Car-Push2 environments, with increases of roughly an order of magnitude in each.

### 4.3.2 REWARD SHAPING RESULTS AND SENSITIVITY TO REWARD SHAPING

The most basic function of the proposed framework is to reduce constraint violations. Figure 2 shows performance/conformance curves in Safety Gym where reward shaping was varied in $\{0, -0.025, -0.005, -0.1, -1\}$. The curves show a meaningful trade-off between the objectives episodic return and cost, which is improved by the use of proposed methods. Table 2 presents the mean and standard deviation of violations per 100 evaluation steps, episode length, and episode reward for reward shaping-enforced constraints with the choice of hyperparameters that produced

Table 4: Mean per-episode MuJoCo rewards and violations with soft dense constraints and constraint state augmentation. Top row displays the reward shaping coefficient $\lambda$.

| Environment | Constraint | Baseline | | Reward Shaping Value | | | | | | | | | |
| | | | | 0 | | −1 | | −10 | | −100 | | −1000 | |
| | | rewards | violations | rewards | violations | rewards | violations | rewards | violations | rewards | violations | rewards | violations |
|---|---|---|---|---|---|---|---|---|---|---|---|---|---|
| Half cheetah | dithering | $1555.30 \pm 27.42$ | $82.84 \pm 6.26$ | $1458.68 \pm 32.23$ | $80.57 \pm 5.74$ | $2054.84 \pm 451.78$ | $73.06 \pm 13.37$ | $\mathbf{2524.10 \pm 436.68}$ | $62.31 \pm 11.25$ | $1495.21 \pm 165.21$ | $43.27 \pm 10.21$ | $639.00 \pm 30.38$ | $\mathbf{16.73 \pm 6.70}$ |
| Reacher | actuation | $-6.55 \pm 0.94$ | $0.61 \pm 0.06$ | $-6.28 \pm 0.51$ | $0.59 \pm 0.04$ | $-6.55 \pm 0.98$ | $0.02 \pm 0.03$ | $\mathbf{-5.28 \pm 0.22}$ | $0.00 \pm 0.00$ | $-8.36 \pm 0.40$ | $0.00 \pm 0.00$ | $-13.44 \pm 0.61$ | $0.00 \pm 0.00$ |
| | dynamic actuation | | $0.00 \pm 0.00$ | $-5.93 \pm 1.67$ | $0.00 \pm 0.00$ | $-5.69 \pm 1.02$ | $0.00 \pm 0.00$ | $-5.53 \pm 1.32$ | $0.00 \pm 0.00$ | $-4.75 \pm 0.88$ | $0.00 \pm 0.00$ | $-11.40 \pm 0.61$ | $0.00 \pm 0.00$ |

Table 5: Atari results with hard constraints, choosing hyperparameters which maximize reward when applying action shaping in training and evaluation, only in training, or only in evaluation.

| Environment | Constraint | Training and Evaluation | | Training Only | | Evaluation Only | |
| | | Mean Episode Reward | Mean Viols/100 Steps | Mean Episode Reward | Mean Viols/100 Steps | Mean Episode Reward | Mean Viols/100 Steps |
|---|---|---|---|---|---|---|---|
| Breakout | actuation | $302.00 \pm 20.75$ | $0.0 \pm 0.0$ | $\mathbf{320.31 \pm 6.09}$ | $0.092 \pm 0.00026$ | $314.91 \pm 13.80$ | $0.0 \pm 0.0$ |
| | dithering | $\mathbf{295.31 \pm 29.07}$ | $0.0 \pm 0.0$ | $276.72 \pm 15.50$ | $0.0073 \pm 3.1e\text{-}05$ | $275.25 \pm 12.67$ | $0.0 \pm 0.0$ |
| | paddle ball | $218.14 \pm 22.85$ | $0.0 \pm 0.0$ | $\mathbf{281.77 \pm 11.03}$ | $0.11 \pm 0.00029$ | $229.00 \pm 11.65$ | $0.0 \pm 0.0$ |
| Seaquest | actuation | $\mathbf{1926.97 \pm 430.24}$ | $0.0 \pm 0.0$ | $1899.78 \pm 502.27$ | $8.30 \pm 0.043$ | $1895.77 \pm 366.79$ | $0.0 \pm 0.0$ |
| | dithering | $\mathbf{2284.78 \pm 15.45}$ | $0.0 \pm 0.0$ | $2256.06 \pm 30.53$ | $0.01 \pm 2.8e\text{-}05$ | $2267.53 \pm 23.94$ | $0.0 \pm 0.0$ |
| SpaceInvaders | actuation | $\mathbf{586.66 \pm 58.69}$ | $0.0 \pm 0.0$ | $582.79 \pm 51.47$ | $14.62 \pm 0.012$ | $583.13 \pm 60.60$ | $0.0 \pm 0.0$ |
| | dangerzone | $613.61 \pm 24.05$ | $0.0 \pm 0.0$ | $\mathbf{733.52 \pm 16.95}$ | $0.0 \pm 0.0$ | $613.82 \pm 27.52$ | $0.0 \pm 0.0$ |
| | dithering | $\mathbf{627.40 \pm 31.43}$ | $0.0 \pm 0.0$ | $624.33 \pm 25.27$ | $0.008 \pm 3e\text{-}05$ | $626.59 \pm 31.04$ | $0.0 \pm 0.0$ |

the minimum violations for each environment/constraint pair in evaluation. We note that the highest value of reward shaping available is generally the best choice for minimizing constraint violations, which were often reduced by an order of magnitude or more from the baseline. Minimizing constraint violations has a small deleterious effect on mean episode reward, but because mean reward per step didn't decrease, episodes were shorter as a result of constraint enforcement.

In addition to minimizing constraint violations, we found that the application of soft constraints can also increase reward per episode. Table 3 presents results for soft constraints with the choice of hyperparameters that produced the maximum reward in each environment/constraint pair. In this case, lower reward shaping values perform best. The hyperparameter values that minimized constraint violations with the Breakout actuation and paddle ball constraints also maximized reward, implying that the objectives were correlated under those constraints. Table 4 presents results for soft constraints with constraint state augmentation in three MuJoCo environments. We find, similar to Atari, that there is one value of reward shaping that is most effective in each environment/constraint pair and that reward degrades smoothly as is shifted from the optimal value.

### 4.3.3 HARD ACTION SHAPING RESULTS

Table 5 presents results for hard constraints with the hyperparameters that produced the maximum return for each environment/constraint pair. Results for cases where hard action shaping was only applied during training or only applied during evaluation are presented as well. There is a slight trend indicating that using action shaping at train time in addition to evaluation increases performance. For those constraints that are qualitatively observed to constrain adaptive behavior, performance rises when using hard shaping only in training, at the cost of allowing constraint violations.

## 5 DISCUSSION

The ability to specify MDP constraints in formal languages opens the possibility for using model checking (Kupferman et al., 2000; Bouajjani et al., 1997), agnostic to the choice of learning algorithm, to verify properties of a safety constraint. Formal language constraints might be learned from exploration, given a pre-specified safety objective, and, because of their explicitness, used without complication for downstream applications or verification. This makes formal language constraints particularly useful in multi-component, contract-based software systems (Meyer, 1992), where one or more components is learned using the MDP formalism.

Experiments with more complex constraints are necessary to explore yet unaddressed challenges, the primary challenge being that the constraints used with action shaping in this work ensured the allowed set of actions was never empty. If this is not the case, lookahead might be required to guarantee zero constraint violations. Further, the tested hard constraints were only used with DQN, which provides a ranked choice over discrete actions. Future work might investigate how to choose optimal actions which are not the first choice in the absence of ranked choice.

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
