# OpenReview forum: "Formal Language Constrained Markov Decision Processes"
_ICLR.cc/2021/Conference — Reject_

### Official Review · AnonReviewer4 · 2020-10-25
**Novel idea with encouraging results, concerns re: scalability and some clarity issues**

**Rating:** 6
**Confidence:** 3

**Review:**

The paper builds on the constrained MDP framework (Altman, 1999), by considering the special-case where the cost functions are defined in terms of states from a parser of a formal language. In the experiments the work uses deterministic finite automata (DFA) but in principle other more expressive classes could be used. Using a formal language to specify constraints may simplify model checking (although this is left to future work).

Pros:
  - Novel idea: not aware of any other work using formal languages for MDP constrained. Although as the paper notes, formal languages (especially those derived from LTL formulae) have been used elsewhere in RL.
  - Some encouraging empirical results: outperforms baseline in safety Gym, and is able to get higher episode return (!) as well as lower violations in some MuJoCo and Atari environments. (The effect of higher returns likely disappears with longer training or more powerful RL algorithms -- but the constraints being a useful inductive bias is nice.)

Cons:
  - While the paper shows that useful constraints *can* be expressed in formal languages, it does not demonstrate that formal languages are a useful way of expressing the constraints. All the constraints in section 4.1 could have been specified quite easily by other means -- e.g. directly writing a cost function in Python. There is vague discussion in section 1 & 5 about how formal languages could enable model checking or connect more generally with safety engineering, but I'm unconvinced.

   In particular, while formal languages might make for a specification that is amenable to verification, the neural network policy that is learned will still be difficult to verify -- especially when the properties depend on the (unknown) transition dynamics as they often do. Moreover, there is no reason why the constraints used in training need to be the same as those used for checking. Why not use a more expressive set of constraints in training, and then check the subset we can easily specify in a formal language?
  - The method requires hand-designing appropriate constraints. In the experiments these were different for each class of environments (safety Gym, MuJoCo, Atari) and in some cases were specific to the environment (e.g. "paddle-ball" and "danger-zone"). I'm sympathetic to this -- safe exploration must need some extra source of information besides just task reward -- but given this there needs to be some discussion or ideally evaluation of the usability of this procedure. It certainly seems intractable for a normal end-user (unlike methods based on e.g. learning from demonstrations or preference comparisons).

  Moreover, it seems quite challenging even for experienced RL practitioners to scale into complex environments. For example, the Atari constraints require having access to some high-level representation of the environment (paddle position, ball position). In real-world robotics, an analogue of this would require training a real-time object localization system.
  - The clarity of the submission could be improved in places. In particular, a clear up-front definition of the problem setting would be useful. Are the transition dynamics known? Unknown? Not known analytically but with oracle query access? Are the cost functions Markovian or not?
  - Hard action space seems to require a resettable environment or known transition dynamics, which weakens the results. RL usually assumes unknown transition dynamics: you should compare to baselines that also make use of this information for fairness, or at least make explicit this difference.

I find the paper borderline -- the idea is interesting and the results encouraging -- but the serious open questions re: whether formal languages are useful and, if they are, whether the approach is scalable make me vote against acceptance.

Some particular things I would suggest addressing in revisions:
  1. Greater justification of why formal languages. This is the most important point here and I'd consider increasing my vote if the usefulness of this approach was clarified. For example, can you cite any papers that use formal languages as specification to validate black-box systems (like RL policy + unknown MDP)?
  2. Discussion of how this method can scale, ideally with empirical validation. For example, could you actually learn a formal language specification (probably hard but impressive), or at least have a case-study where you apply this approach to something that doesn't have a nice low-dimensional state representation? Alternatively, if this method is limited to low-dimensional environments -- why use your method here and not a more classical control method?

Some clarifying questions I'd also appreciate if the authors could address during the discussion:
  1. How is $t_v$ computed? It seems underdefined -- it is a function of $q_t$ but the natural language definition given seems like it is in terms of constraint violations from rollouts of the current MDP state. As I understand it, many different states $s_t$ could lead to the same $q_t$ -- nor is making it a function of $s_t$ sufficient since the same $s_t$ could lead to different $q_t$ depending on previous states!

    Even if this issue is resolved (e.g. making it a function of both $s_t$ and $q_t$), computing it seems like it would require rollouts from $s_t$. This seems very expensive -- is my understanding correct? If so, could you report wall-clock times, or possibly train different algorithms for a fixed wall-clock time? Otherwise comparing algorithms for the same number of training timesteps seems unfair if the shaping effectively involves extra hidden environment interactions.
  2. Perhaps related to this confusion, you state in the hard action shaping section, that "before finalizing that choice, simulates stepping the DFA with the resulting token from the translation function". I am confused how one does this -- isn't the point we don't know the transition dynamics, so do not know the next MDP state we will end up in? If we do have a transition model, why don't we do planning instead to never take a sequence of actions that violate constraints?
  3. Any ideas why hard action shaping in both Training and Evaluation sometimes does worse than just during Training or just during Evaluation?

Some more detailed suggestions for improvements (no need to respond to these, they did not affect my decision in any significant way):
  - Page 2, “teacher advice”. “RL safety” is not a synonym for “safe exploration”, since there are other forms of safety related to RL (e.g. satisfying constraints in the final policy; optimizing a desirable objective, like user happiness, rather than a misaligned one, like short-term user engagement).
  - Page 3, \delta: what is $A$? It is not defined anywhere that I can see – should this be $\Sigma$?
  - Page 3, $D_C$ is not a function since it takes only a single token but outputs the next state, which is dependent on the current state (not supplied). This also makes the constraints non-Markovian which seems like it should be emphasized more (although this is somewhat implicit in the “Constraint State Augmentation” section).
  - Page 3, Constraint State Augmentation: I think the dimensions should be integer ceiling not integer floor.
  - Page 3, Learned Dense Cost: perhaps make $t_{v}$ upper case since it's a random variable. This method also seems a bit ad-hoc – is there a theoretical justification for it? If not, how do other choices (e.g. a different exponent base to 1/2) change things?
  - Page 7-8: tables of numbers were a bit hard to digest. Consider turning some of them into figures, e.g. bar charts?
  - Table 5: a bold entry for “Training Only” in Breakout seems incorrect (for dithering, “Training and Evaluation” does better).
  - Discussion: “was never be empty”->”was never empty”. (“Best effort” also seems an odd phrase for this property.)

### Update: the author's revision have clarified many of the points of confusion above, and have largely addressed my concern re: the value of formal languages in this setting. I continue to be concerned about the scalability of this class of methods, but since it is a problem common to other work in this field, I do not want to hold this too much against the current submission. Given this I have increased my score to a 6. ###

---

> ### Author Response · Authors · 2020-11-24
> **Author Response**
>
> Q. Why not use a more expressive set of constraints in training, and then check the subset we can easily specify in a formal language?
>
> A. In principle, there is no reason why someone using methods similar to ours couldn't do exactly that. We had two main reasons for using formal language constraints. First, we are interested in being able to provide probabilistic guarantees on learned policies, with methods which are well known in software engineering and cyberphysical systems, which are usable downstream in larger, multi-component software systems and compatible with well known methods in contract-based formal verification. E.g., De Miguel et al. [0], which discusses the use of formal safety languages in integrating software engineering and safety engineering, Hatcliff et al. [1] and Armstrong et al. [2], which survey behavioural interface specification languages with a focus toward automatic program verification; Moy and Marche [3], which automatically infers and composes safety contracts from formal specifications of safety in software components; and Aziz et al.~[4] and Wasilewski et al. [5], which discuss the uses and the benefits of a family of formal languages called Domain Specific Modeling Languages, used in modelling cyberphysical systems. The second reason is that the automata which encode formal language constraints have a useful state and we found satisfactory empirical performance in our experiments using this state in augmenting MDP state for representation learning in the policy as well as reward/cost shaping.
>
> With respect to the problem of verifying RL policies, progress has already been made in verifying neural network-represented policies in cyberphysical systems (e.g., Tran et al. [6]), and formal languages are one way to specify checkable constraints with existing neural network verification methods. The problem of handling unknown MDP dynamics is a significant one in the RL verification problem, but more work is required in the field to tackle this much larger problem.
>
> [0] Miguel A De Miguel, Javier Ferńandez Briones, Juan Pedro Silva, and Alejandro Alonso.  Integration of safety analysis in model-driven software development. IET software, 2(3):260–280, 2008.
>
> [1]  John  Hatcliff,  Gary  T  Leavens,  K  Rustan  M  Leino,  Peter  Muller,  and Matthew Parkinson.   Behavioral  interface  specification  languages. ACM Computing Surveys (CSUR), 44(3):1–58, 2012.
>
> [2]  Robert C Armstrong, Ratish J Punnoose, Matthew H Wong, and Jackson R Mayo.  Survey of existing tools for formal verification. SANDIA REPORTS 2014-20533, 2014.
>
> [3]  Yannick Moy and Claude Marche.  Modular inference of subprogram contracts for safety checking. Journal of Symbolic Computation, 45(11):1184–1211, 2010.
>
> [4]  Muhammad Waqar Aziz and Muhammad Rashid. Domain specific modeling language for cyber physical systems.  In 2016 International Conference onInformation Systems Engineering (ICISE), pages 29–33. IEEE, 2016.
>
> [5]  Michael Wasilewski, Wilhelm Hasselbring, and Dirk Nowotka.  Defining requirements on domain-specific languages in model-driven software engineering  of  safety-critical  systems. Software Engineering 2013 Workshop,2013.
>
> [6]  Hoang-Dung  Tran,  Feiyang  Cai,  Manzanas  Lopez  Diego,  Patrick  Musau, Taylor T Johnson,  and Xenofon Koutsoukos.  Safety verification of cyber-physical systems with reinforcement learning control. ACM Transactions on Embedded Computing Systems (TECS), 18(5s):1–22, 2019.
>
> Q. Discussion of how this method can scale
>
> A. We expect that the proposed methods can scale to very challenging environments and complex constraints as it makes no assumptions about the size/complexity of the underlying MDP or the size/complexity of the constraint. The use of relatively high-level features as input to the constraint automata is a common feature of many experiments found in work on constraints, so the challenge of scaling constraints is one that is not unique to our work. We aren't familiar with any existing work which learns a constraint from low-level features.
>
> We are looking forward to future work in learning a translation function to apply constraints to environments in which high-level features are more difficult to capture. Because this is a significant challenge in its own respect, we believe this direction is beyond the scope of the present work and would be more appropriately studied in its own paper.

---

> > ### Author Response · Authors · 2020-11-24
> > **Author Response Pt. 2**
> >
> > Q. The clarity of the submission could be improved in places. In particular, a clear up-front definition of the problem setting would be useful. Are the transition dynamics known? Unknown? Not known analytically but with oracle query access? Are the cost functions Markovian or not?
> >
> > A. We assume that transition dynamics are unknown and that exploration (we use epsilon greedy in our experiments) must be used to learn a policy. The cost functions may be non-Markovian in their specification, but the inclusion of the automata state in the MDP state makes the representation of any constraint Markovian, simliar to what is explored in Camacho et al. [1], which studies the solution of non-Markovian Reward Decision Processes with MDP solvers by a transformation using automata. We'll clarify this in the paper.
> >
> > [1] Alberto  Camacho,  Oscar  Chen,  Scott  Sanner,  and  Sheila  A  McIlraith. Decision-making  with non-Markovian rewards:  From LTL to automata-based reward shaping.  In Proceedings of the Multi-disciplinary Conference on Reinforcement Learning and Decision Making (RLDM), pp.279–283, 2017b.
> >
> > Q. Hard action space seems to require a resettable environment or known transition dynamics, which weakens the results. RL usually assumes unknown transition dynamics: you should compare to baselines that also make use of this information for fairness, or at least make explicit this difference.
> >
> > A. We do not use known transition dynamics in our experiments, though we do assume that an environment is either resettable or has access to a fall-back known-safe policy that can take over in the event that the main policy is unsafe. The latter assumption is not an uncommon one in shielding type methods (e.g., Mao et al. [1] uses such an assumption)
> >
> > [1] Hongzi Mao,  Malte Schwarzkopf,  Hao He,  and Mohammad Alizadeh.  Towards safe online reinforcement learning in computer systems. In 33rd Conference on Neural Information Processing Systems (NeurIPS 2019), 2019.
> >
> > Q. Greater justification of why formal languages. This is the most important point here and I'd consider increasing my vote if the usefulness of this approach was clarified. For example, can you cite any papers that use formal languages as specification to validate black-box systems (like RL policy + unknown MDP)?
> >
> > A. In addition to the papers cited earlier in this response, there are many other works which use formal languages as a safety specification on Markov decision processes. Baier et al. [1] introduces an LTL-based formal language called ProbMela and does probabilistic model checking on MDPs with known dynamics. Zhang et al. [2] uses an LTL-based formal language to specify constraints on cyberphysical systems and a method of checking those constraints in adaptive systems. Brazdil et al. [3] discusses verification of MDPs with qualitative PECTL* objectives, another type of formal language. Bouchekir and Boukala [4] propose a probabilistic symbolic compositional verification approach to verify probabilistic systems where each component is a Markov decision process. Tran et al. [5] verify RL policies represented with neural networks.
> >
> > All of these listed papers assume known MDP dynamics. However, as discussed above, verification of MDPs with unknown dynamics is an open problem. Even without full verification, however, we believe that the use of verification methods in reducing constraint violations is a significant line of work, as discussed in Ray et al. [6].
> >
> > [1]  Christel  Baier,  Frank  Ciesinski,  and  Marcus  Großer.   Probmela  and  verification  of  markov  decision  processes. ACM SIGMETRICS Performance Evaluation Review, 32(4):22–27, 2005.
> >
> > [2]  Ji Zhang, Heather J Goldsby, and Betty HC Cheng. Modular verification of dynamically adaptive systems. In Proceedings of the 8th ACM international conference on Aspect-oriented software development, pages 161–172, 2009.
> >
> > [3]  Tomas Brazdil,  Vojtech Forejt,  and Antonin Kucera.  Controller synthesis and  verification  for  markov  decision  processes  with  qualitative  branchingtime objectives.  In International Colloquium on Automata, Languages, and Programming, pages 148–159. Springer, 2008.
> >
> > [4]  Redouane  Bouchekir  and  Mohand  Cherif  Boukala.   Learning-based  symbolic assume-guarantee reasoning for markov decision process by using interval markov process. Innovations in Systems and Software Engineering, 14(3):229–244, 2018.
> >
> > [5]  Hoang-Dung  Tran,  Feiyang  Cai,  Manzanas  Lopez  Diego,  Patrick  Musau, Taylor T Johnson,  and Xenofon Koutsoukos.  Safety verification of cyber-physical systems with reinforcement learning control. ACM Transactions on Embedded Computing Systems (TECS), 18(5s):1–22, 2019.
> >
> > [6]  Alex  Ray,  Joshua  Achiam,  and  Dario  Amodei. Benchmarking  safe  exploration  in  deep  reinforcement  learning. arXiv preprint arXiv:1910.01708,2019.

---

> > > ### Author Response · Authors · 2020-11-24
> > > **Author Response Pt. 3**
> > >
> > > Q. How is $t_v$ computed?
> > >
> > > A. Similar to how the Lagrangian penalty parameter is calculated, we calculate  an  empirical  rolling  average  (essentially a Monte  Carlo estimate  over trajectories) for each constraint state. It is true that many different trajectores can lead to the same $q_t$, but the estimation can be agnostic to the particular MDP state and trajectory because the constraint state is Markovian and is calculating an expected value over trajectories. It's assumed that the constraint state alone is sufficient to estimate a reasonable average over trajectories, especially for a fixed or slowly changing policy. Our experiments do not use any more rollouts than the typical RL training loop or the methods we compare against.
> > >
> > > Q. How do we simulate a DFA step?
> > >
> > > A. We only perform this lookahead in state-independent constraints in our experiments. This is a weakness of our method that we propose the use of the methods of Dalal et al. [1] to remedy, which transforms state-based constraints into action-based constraints by learning a linear approximation of the state and action-conditional cost function. We will be implementing exactly this method in future work.
> > >
> > > [1] G. Dalal, K. Dvijotham, M. Vecerik, T. Hester, C. Paduraru, and Y. Tassa. Safe Exploration in Continuous Action Spaces. 2018.
> > >
> > > Q. Any ideas why hard action shaping in both Training and Evaluation sometimes does worse than just during Training or just during Evaluation?
> > >
> > > A. Our hypothesis is that when a constraint is too restrictive for an environment, it will perform better when not employed at test time, and will perform better when not employed at train time when useful intermediate policies (i.e., those that are employed in training but are different from the policy at the end of training) are forbidden by the constraint. A potential direction of future work might be to study when hard, action-shaping constraints would be expected to help or hurt performance by their effect on exploration and the final learned policy.

---

> > > > ### Comment · AnonReviewer4 · 2020-11-25
> > > > **Thanks for the clarification**
> > > >
> > > > Thank you for the detailed response. This has addressed some of my concerns. In particular, I better understand the value of using formal languages in this setting.  I'm still not convinced about the method's scalability given the requirement for hand-designed constraints but, as you point out, this is a more general problem in this domain. Given this, I'll increase my score.

---

### Official Review · AnonReviewer2 · 2020-10-27
**Novel design strategy for constrained RL**

**Rating:** 6
**Confidence:** 3

**Review:**

#### Summary
The authors propose to use formal languages, specifically DFAs, as a mechanism to specify constraints in a constrained MDP setting. This has the benefit of being able to rely on a large body of existing work on identification, safety verification, etc. The strategy relies on decomposing the constraint into a translation, recogniser & cost assignment function that connect the MDP to the DFA. The mentioned cost can then be combined with existing solution for solving cMDPs, such as reward shaping and Lagrangian methods. The key observation is that adding the recogniser state to the observations of the policy can result in significant gains in both performance and constraint satisfaction. A range of results are presented across different environment suites and hyper parameters.

#### Pros
- A novel design strategy for specifying constraints in constrained MDPs, which have become very popular again as means of learning safe controllers in e.g. robotics.
- Extensive evaluation on both discrete and continuous domains with various constraints, optimised hypers and multiple seeds.
- Results are largely in favour of the proposed method, esp. the benefit of adding the constraint state as an observation.
- Very clear and concise presentation (except minor comments, see below).

#### Cons
- The significance of this work is perhaps lower. While using the framework of formal languages to define constraints is a novel design strategy, the methods employed in this paper to resolve the resulting cMDP are not novel beyond adding the constraint state to the policy's observations. Any of the mentioned benefits of using formal languages, such as verification, are not actually investigated. I also would have liked to see more exploration of the benefit / shape of the dense cost function, as this now gets lost as a binary hyper parameter in the tables.
- The chosen constraints are not very interesting as they pertain largely to sequences of actions only (except for Safety Gym). Even for what would intuitively be state-dependent constraints such as paddle ball distance, the authors specify the translation function in such a way that it becomes an action-only constraint. This makes action shaping significantly easier.

#### Comments
- I couldn't find the definition of the cost assignment functions for the environments. Are they just binary?
- When a constraint gets violated, the DFA gets into an accepting state but it seems the episode does not get terminated. Does the DFA get reset in the next step?
- Using "reward shaping", "cost shaping" and "dense reward" can be confusing as they're often used interchangeably in other works.
- Figure 2 is not discussed in the text at all. Do the data points represent different values for $\lambda$?
- In Table 3 seaquest actuation and Table breakout dithering the wrong result is marked in bold (indicating best performance).

#### Conclusion
Overall, even though the significance is perhaps limited, I vote to borderline accept this paper due to the clarity and thorough evaluation.

---

> ### Author Response · Authors · 2020-11-24
> **Author Response**
>
> Q. I couldn't find the definition of the cost assignment functions for the environments. Are they just binary?
>
> A. Yes, in all environments the cost function was binary. Binary cost is built into the Safety Gym domain and we used binary cost elsewhere for consistency. We'll note this in the paper.
>
> Q. When a constraint gets violated, the DFA gets into an accepting state but it seems the episode does not get terminated. Does the DFA get reset in the next step?
>
> A. Yes it does. Note that in state-based constraints, like Safety Gym, the constraint can be violated in consecutive timesteps if no action has been taken to move out of the forbidden state (i.e., the DFA is reset but in the next step transitions into a violating state again).
>
> Q. Figure 2 is not discussed in the text at all. Do the data points represent different values for $\lambda$?
>
> A. Yes, the data points in Fig. 2 represent different values for $\lambda$, the reward shaping coefficient. The text discussing it which was accidentally omitted from Sec. 4.3 will be posted in a separate thread. Apologies.
>
> To address the first con listed, the significance of this work is partly in studying how to bring methods from cyberphysical systems and software engineering (i.e., formal language constraints) together with RL methods (i.e., the CMDP formalism) in a way that works well with modern RL algorithms and that solves challenging environments proposed by Ray et al. [1]. Though the added benefits of formal languages like verification are not studied in this paper, they are an important part of the motivation to study the use of formal language constraints in RL. Future work aside, we believe the clear improvements in empirical performance in Safety Gym environments proposed by Ray et al. clearly demonstrates the benefits of using the specific formal language CMDP construction we proposed along with the state augmentation it enables (along with the benefits in Mujoco and Atari environments).
>
> [1] Alex Ray, Joshua Achiam, and Dario Amodei. Benchmarking safe exploration in deep reinforcement learning. arXiv preprint arXiv:1910.01708, 2019

---

### Official Review · AnonReviewer3 · 2020-10-29
**Unclear contribution and lack of contrast and comparison with relevant literature**

**Rating:** 5
**Confidence:** 5

**Review:**

#### Summary
The paper proposes a constrained reinforcement learning (RL) formulation relying on constraints written in a formal language. The proposed formulation is based on constrained Markov decision processes where the constraint is represented as a deterministic finite automaton that rejects any trajectory violating the constraint. The proposed solution relies on transforming the automaton's sparse binary cost into an approximate dense cost and augmenting that with the reward objective. The paper presents a series of results from simulations in Safety Gym, MuJoCo, and Atari environments.

#### Strength
1. Constrained RL is certainly an important research area, having a variety of applications such as safety-critical problems.
2. The constraints written in a formal language can represent structured properties, including non-Markovian ones.
3. While a constraint in a formal language requires some domain knowledge, if it is correct, it can accelerate the training phase.

#### Weakness
1. The paper cites some of the related works; however, it is unable to distinguish its contributions compared to the existing methods. In particular, some of the most relevant approaches are using linear temporal logic (LTL) formulas for reward shaping (Camacho et al., 2017a;b), using LTL constraints in RL (Hasanbeig et al., 2018), and shielding mechanisms (Jansen et al., 2018; Alshiekh et al., 2018). The paper cites these references but does not differentiate itself from them.
2. The experimental results of the paper are limited in terms of comparison with the existing methods. In particular, the only comparison is with baselines that do not use this automaton-based side information. However, since there exist many works with the capability of using such side information (some are mentioned in the previous point), these comparisons are essential for correctly evaluating the paper’s contribution and significance.

#### Recommended Decision
Given the paper's unclear contribution and lack of necessary comparison with the existing literature, I recommend rejecting the paper in its current form.

#### Supporting Arguments
1. The paper motivates the automaton-based constraints mostly for safety-critical applications. Nonetheless, the soft version of the constraints still lead to constraint violations and provide no safety guarantees.
2. In addition to citing the related work, the paper needs to clearly state the differences with the existing methods.
3. The writing requires somewhat considerable polishing. There are many typos, grammatical errors, and awkward phrases. Some notations are undefined, e.g., state space $S$ and action space $A$, $\tau$, operators in regular expressions, alphabets $n$ and $f$ in Figure 1(b), and the evaluation metric called accumulated cost regret. While the reader can probably infer some of them, the paper should be self-contained in this regard.
4. Figure 2 is not mentioned nor discussed in the paper. Some figures and tables are inserted far from where they are initially mentioned.
5. To properly answer the second question posed in Section 4.3 regarding the choice of hyperparameters, a criterion, strategy, or heuristic is required. Reporting the better hyperparameters for some instances, on its own, is not sufficient as an answer.

#### Questions
1. How is $t_v(q_t)$ estimated using rollouts, given that the framework and the simulations seem to be for the model-free setting?
2. Could you please explain the reasoning for the choice of $t_v^{baseline}$?

#### Additional Feedback for Improving the Paper
1. The following references are also highly related to this paper:
    - Zhu, He, et al. "An inductive synthesis framework for verifiable reinforcement learning." Proceedings of the 40th ACM SIGPLAN Conference on Programming Language Design and Implementation. 2019.
    - Fulton, Nathan, and André Platzer. "Verifiably safe off-model reinforcement learning." International Conference on Tools and Algorithms for the Construction and Analysis of Systems. Springer, Cham, 2019.
2. Compound adjectives require hyphens, e.g., “formal language constrained MDP” $\to$ “formal-language-constrained MDP”.
3. The clarity of the paragraphs on “Hard Constraints and Action Shaping” can be improved.
4. Please further motivate and explain why the constraints introduced in the simulation setting (except “proximity”) are required.
5. Please make the bibliography consistent in terms of details provided for each reference and the formatting. Also, the year of some references is repeated.

---
#### UPDATE
I thank the authors for their response. I am still concerned about the paper's novelty and contribution compared to the existing work as the differences seem minor. Also, the comparison with the approach by Camacho et al. (2017a;b) is not sufficient since different values of hyperparameter $\lambda$ may perform better. Given the authors' revisions and the added results, I have increased my score. However, I am still inclined toward rejecting the paper.

---

> ### Author Response · Authors · 2020-11-24
> **Author Response**
>
> We'll add the differences between our work and the works listed to the related work section of the paper, but we'll list them here as well:
>
> - Camacho et al. (2017a;b) both focus on the problem of using automata to represent non-Markovian reward functions and reward shaping to train the policy. Our work uses the CMDP formalism and cost function, and shapes the cost function with a learned rather than static potential function on the automata states. Further, our work augments the MDP observation with the automata state and Camacho et al. (2017a;b) do not. Further, our framework strictly generalises the methods in these papers: We can recreate their methods by using reward shaping as the CMDP mechanism and a particular choice of dense cost shaping which is not learned, as well as not using state augmentation in policy learning. We'll post results of our experiments with the method described in Camacho et al. 2017b on Safety Gym environments below.
>
> - Hasanbeig et al.  (2018) forms a product MDP to avoid constraint violations defined in LTL, but approximates the Q-function with one network per automaton state and defines a reward function which only returns a positive value when the automaton transitions to an accepting state. Our work doesn't prescribe anything about the form of the learner other than having an input for the automaton state, or the MDP reward function (e.g., when we use a Lagrangian method rather than reward shaping). We do not compare their methods to ours experimentally because they study behavioural constraints which must be satisfied, rather than avoided, and define the reward function accordingly. It is unclear how the reward function they define should be adapted to the case of avoiding constraint violations.
>
> - Jansen et al. (2018) and Alshiekh et al. (2018) each construct a probabilistic shield to avoid unsafe behavior similar to our hard constraints and action shaping. However, each requires the underlying MDP states to be enumerable and few enough so that a shield can be constructed efficiently, which our hard shaping does not need. Further, neither explore the addition of constraint state to the MDP state for learning. We do not compare their methods to ours experimentally because they are intractable in the environments used in our experiments.
>
> Q. How is $t_v$ estimated using rollouts, given that the framework and the simulations seem to be for the model-free setting?
>
> A. Similar to how the Lagrangian penalty parameter is calculated, we calculate an empirical rolling average (essentially a Monte Carlo estimate over trajectories) for each constraint state.
>
> Q. Reasoning for choice of $t_v^{baseline}$
>
> A. Similar to Ray et al. (2019) [1], we set a target for how many violations per 1000 steps are allowable. This target is used both in the calculation of the Lagrangian penalty parameter and the cost shaping function we propose. Its effect in the equation is to make the exponent close to 1 when the empirical $t_v$ is close to the target and make the exponent greater than or less than 1 exactly when $t_v$ is greater than or less than the target.
>
> [1] Alex Ray, Joshua Achiam, and Dario Amodei. Benchmarking safe exploration in deep reinforcement
> learning. arXiv preprint arXiv:1910.01708, 2019.
>
> Q. Please further motivate and explain why the constraints introduced in the simulation setting (except “proximity”) are required.
>
> A. Constraints which model undesirable, but not catastrophic,  properties of a robotic policy, e.g., overactuation of a motor with "actuation" and high frequency jitter with "dithering", were introduced in Mujoco as a model robotics environment. These same constraints are applied to Atari to see if simple, action-based constraints could help guide policy learning to higher reward outcomes as a form of teacher advice. The "paddle ball" and "dangerzone" constraints were designed for their Atari environment specifically to see if the same effect could be achieved with a weak notion of safety (i.e., avoiding states which end the episode).
>
> Thank you for bringing the papers you listed to our attention. We'll add them to the related work.

---

> > ### Author Response · Authors · 2020-11-24
> > **Author Response (Camacho et al. experiment results)**
> >
> > The method of Camacho et al. (2017b), uses reward shaping with a potential function such that the potential of each automaton state grows linearly as the distance to an accepting state decreases. Originally, they use it for specifying objectives, but there is nothing essential about the method that keeps it from being used to specify constraints instead. Our analysis is that, aside from in the Point Goal1 environment, the lack of flexibility in the shaping leads to policies which are extremely conservative. Qualitatively, the learned policy doesn't move hardly at all, seemingly in order to avoid potentially violating the constraint. This problem can be ameliorated either by not using shaping and only penalising when an accepting state is reached (allowing for more flexibility in the policy when not in close proximity to violating states), or learning the shaping function as we propose, in order to correctly assign very low cost to moving into automaton states which are found to be safe in training.
> >
> > These are the results of running Camacho et al. (2017b) in Safety Gym environments with reward shaping coefficient -0.001 (equal to the lowest value we employ in our methods). The values have been normalised with the same baselines as the results of Table 1 in the submission. More results will be posted as they finish, as a last few experiments are still running.
> >
> > Point Goal 1
> > Return:		 0.3141
> > Violation:	 0.0928
> > Cost Rate:	 0.1037
> >
> > Point Goal 2
> > Return:		-0.0193
> > Violation:	 0.1799
> > Cost Rate:	 0.2404
> >
> > Point Button 1
> > Return:		-0.0086
> > Violation:	 0.0463
> > Cost Rate:	 0.102
> >
> > Point Button 2
> > Return:		-0.0578
> > Violation:	 0.0547
> > Cost Rate:	 0.016
> >
> > Point Push 1
> > Return:		-0.0062
> > Violation:	 0.0779
> > Cost Rate:	 0.0690
> >
> > Point Push 2
> > Return:		-0.9914
> > Violation:	 0.0963
> > Cost Rate:	 0.079
> >
> > Car Goal 1
> > Return:		-0.0031
> > Violation:	 0.154
> > Cost Rate:	 0.1919
> >
> > Car Goal 2 Return: 0.0021 Violation: 0.0715	Cost Rate: 0.0872
> >
> > Car Button 1 Return: 0.0252 Violation: 0.1418 Cost Rate: 0.0494
> >
> > Car Button 2 Return: -0.0083 Violation: 0.0757 Cost Rate: 0.087
> >
> > Car Push 1 Return: 0.0085 Violation: 0.0581 Cost Rate: 0.0722
> >
> > Car Push 2  Return: 0.096 Violation: 0.0493 Cost Rate: 0.0231
> >
> > edit: Added the rest of the experiments in Car Goal2, Car Button1,2, Car Push1,2

---

### Official Review · AnonReviewer1 · 2020-10-29
**Novelty and technical contribution of the paper is unclear.**

**Rating:** 6
**Confidence:** 5

**Review:**

update after rebuttal: I think the paper, with the added discussion, improved.

Summary and Contribution

This paper concerns safe reinforcement learning. In particular, it takes the perspective of constraining system behavior using formal languages instead of the usual constraint MDP framework where an additional (simple) cost function is used. The constraints are given as finite automata which are basically used as a product with the original MDP state space. The paper provides an empirical evaluation using several state of the art benchmarks.



Reasons for Score

With more details on the correctness, better literature study, and more experiments, the paper would be a clear accept. As of now, I see it as marginally above the acceptance threshold.

Strengths

- A relevant problem is tackled.
- The usage of formal languages is relevant and has been demonstrated to be helpful in a number of results

Weaknesses

- The contribution of the paper is not clear. Several other works have done very similar approaches, and in much more depth.
- The assumptions are not clear.

Questions for Authors

- Please compare your work to the literature listed below. What is novel, how does it relate?
- What is the exact assumption on prior knowledge? Does the MDP need to be known beforehand, or is the exploration of an RL algorithm basically guided by the finite automaton?
- In the construction of the formal setting, what is different to the standard product construction of MDPs and finite automata?


Detailed Comments

- related work

At least three papers (one of them is cited) constrain MDP exploration using regular languages (in fact, omega-regular languages). I don't see the novelty, but I'm happy to be convinced otherwise. See:

Mohammadhosein Hasanbeig, Alessandro Abate, Daniel Kroening:
Logically-Correct Reinforcement Learning. CoRR abs/1801.08099 (2018)

Mohammadhosein Hasanbeig, Alessandro Abate, Daniel Kroening:
Cautious Reinforcement Learning with Logical Constraints. AAMAS 2020: 483-491

Ernst Moritz Hahn, Mateo Perez, Sven Schewe, Fabio Somenzi, Ashutosh Trivedi, Dominik Wojtczak:
Omega-Regular Objectives in Model-Free Reinforcement Learning. TACAS (1) 2019: 395-412

- formal construction
In the very short theory part of the paper, the construction seems to me just as the standard construction to a product of an MDP and a finite (omega-regular) automaton. In a nutshell, one has a labelled MDP where the labels correspond to the alphabet of an automaton. Then, a run of the MDP produces a sequence of labels (a trace), which can be checked by the automaton. Now, a product construction of both yields an MDP where a reachability computation for so-called end components gives the probability to satisfy the formal language constraint. See for instance text books such as Baier and Katoen, Principles of Model Checking. As this part is the main body of the technical body of the paper, I would really like to understand, what is the difference.

---

> ### Author Response · Authors · 2020-11-24
> **Author Response**
>
> Q. Please compare your work to the literature listed below. What is novel, and how does it relate?
>
> A. The main novelty of our work is the use of the CMDP framework with constraint automata and methods which make the use of the inductive bias of the automaton state with RL policies represented by neural networks (though there is nothing essential about neural networks), as well as experiments which quantify feasibility and utility. The suggested papers are similar to ours in their use of automata encoding constraints and the use of a mechanism to encourage an RL policy to satisfy those constraints, but the mechanisms employed by those papers seem relatively bespoke and limit the previous work to relatively small MDPs. The suggested papers' use of formal languages in RL safety further motivates our work in combining automata and deep learning in a more general way.
>
> The most significant differences between our work and Hasanbeig et al. (2020) are 1) their use of the observation area which requires knowledge of the neighbors in the MDP graph, which is unsuitable for use in large, highly connected, or unknown MDPs (vs. no such requirement) and, 2) their mechanism used to enforce safety constraints ("safe padding" and double learners for them vs. CMDP formalism and cost/reward shaping or Lagrangian method for ours). Hasanbeig et al. (2018) is similar.
>
> Hahn et al.'s goal is to reach an objective almost surely, and uses reward shaping with very large values compared to the underlying reward function to ensure that off-the-shelf model-free RL will satisfy the objective. This strategy does not work in the case of a constraint rather than an objective because it will lead to an overly conservative policy, similar to the case in our experiments where a large reward shaping value leads to both very low constraint violations and return.
>
> Overall, the core ideas are similar, but there are significant problems with efficiency when Hasabeig et al.~(2018,2020) is applied to larger, unknown MDPs, or with applying the objective-specification method of Hahn et al. to specifying a constraint instead.
>
> Q. What is the exact assumption on prior knowledge? Does the MDP need to be known beforehand, or is the exploration of an RL algorithm basically guided by the finite automaton?
>
> A. The proposed setting does not require knowledge of MDP dynamics, and the RL agent must use exploration to learn a policy; epsilon-greedy exploration was employed in our experiments. The finite automaton can, through the cost or reward signals, provide a weak incentive or disincentive to visit certain MDP states, which allows a human with any knowledge of MDP dynamics to provide a weak inductive bias on the exploration (e.g., what action sequences are known to perform poorly a priori and thus should not be used repeatedly in exploration, like violently overactuating a robot limb or dithering by moving back and forth repeatedly).
>
> Q. How is this different from the standard approach described in e.g., the textbook ``Principles of Model Checking''?
>
> A. The core approach of using a product MDP is identical; however, calculating the probability of satisfying the formal language constraint in an MDP as large as we're working in is intractable, so we use optimisation instead to try to find a policy which best satisfies the dual objectives of constraint satisfaction and accumulated reward over a rollout. The novelty in this work isn't in the idea of the product MDP, but rather the use of the CMDP framework with the product MDP in order to bring together well known CMDP mechanisms with formal language constraints, as well as the methods of augmenting the RL policy input with the constraint automaton state and dense cost shaping. I.e., we study how to get the product MDP construction to work well with large MDPs and modern RL algorithms which train policies represented by neural networks. This is demonstrated by the choice of environments in our experiments.

---

### Author Response · Authors · 2020-11-24
**Rebuttal Revision Change Notes**

Thank you for all of the very helpful feedback! To summarise our changes to the submission:

Sec. 2, par. Formal Languages
- Added citations:
	- Zhu, He, et al. "An inductive synthesis framework for verifiable reinforcement learning." Proceedings of the 40th ACM SIGPLAN Conference on Programming Language Design and Implementation. 2019.
	- Fulton, Nathan, and André Platzer. "Verifiably safe off-model reinforcement learning." International Conference on Tools and Algorithms for the Construction and Analysis of Systems. Springer, Cham, 2019.
	- Mohammadhosein Hasanbeig, Alessandro Abate, Daniel Kroening: Cautious Reinforcement Learning with Logical Constraints. AAMAS 2020: 483-491
	- Ernst Moritz Hahn, Mateo Perez, Sven Schewe, Fabio Somenzi, Ashutosh Trivedi, Dominik Wojtczak: Omega-Regular Objectives in Model-Free Reinforcement Learning. TACAS (1) 2019: 395-412

Sec. 2, par. Teacher Advice
- "safe RL" -> "safe exploration"
- Added text differentiating our work from the cited shielding papers, (Jansen et al. 2018) and (Alshiekh et al., 2018), and the papers on solving non-Markovian reward decision processes (Camacho et al., 2017a;b)

Sec. 3:
- Updated the definition of $D_c$ and the CMDP const function $c$ to account for the tracking of the constraint state $Q_c$
- Added a note specifying that no knowledge of the MDP transition function is necessary nor does

Sec. 3, par. Constraint State Augmentation:
- Changed incorrect floor to ceiling

Sec. 3, par. Recognizer Function
- Corrected part of the definition of $\delta$ from "A" -> $\Sigma$

Sec. 3, par. Learned Dense Cost
- $t_v$ -> $T_v$ to properly reflect that it's a random variable
- Added that $T_v$ is calculated as a Monte Carlo estimate of the expected value and is tracked by an exponential moving average.

Sec. 4.1:
- Added a note that the cost functions are binary unless we're performing dense cost shaping.

Sec. 4.3.2:
- Added very brief discussion of Fig. 2 to the text.

Sec. 5:
- “was never be empty” -> ”was never empty”

Tables:
- Fixed bolding

---

### Decision · Program_Chairs · 2021-01-07
**Final Decision**

**Decision:**

Reject

**Comment:**

The paper proposes formulating safety constraints as formal language constrains, as a step toward bridging the gap between ML and software engineering, and enabling safe exploration in RL.  The authors responded and improved the paper significantly during the rebuttal period. Despite that, the reviewers raise the question, and I agree, that the significance of the paper, especially the novelty of the method, do not meet ICLR standard. The future version of the paper should be developed more in terms of the novelty, evaluations, and related works.